# Application of Floating Aquatic Plants in Phytoremediation of Heavy Metals Polluted Water: A Review

**Shafaqat Ali [1,2,*], Zohaib Abbas [1], Muhammad Rizwan [1], Ihsan Elahi Zaheer [1], İlkay Yavaş [3], Aydın Ünay [4], Mohamed M. Abdel-Daim [5,6], May Bin-Jumah [7], Mirza Hasanuzzaman [8] and Dimitris Kalderis [9]**

[1] Department of Environmental Sciences and Engineering, Government College University, Allama Iqbal Road, Faisalabad 38000, Pakistan; shafaqataligill@yahoo.com (S.A.), zohaib.abbas83@gmail.com (Z.A.); mrazi1532@yahoo.com (M.R.); ihsankhanlashari@gmail.com (I.E.Z.)

[2] Department of Biological Sciences and Technology, China Medical University, Taichung 40402, Taiwan

[3] Department of Plant and Animal Production, Kocarli Vocational High School, Aydın Adnan Menderes University, 09100Aydın, Turkey; iyavas@adu.edu.tr

[4] Department of Field Crops, Faculty of Agriculture, Aydın Adnan Menderes University, 09100 Aydın, Turkey; aunay@adu.edu.tr

[5] Department of Zoology, College of Science, King Saud University, P.O. Box 2455, Riyadh 11451, Saudi Arabia; abdeldaim.m@vet.suez.edu.eg

[6] Pharmacology Department, Faculty of Veterinary Medicine, Suez Canal University, Ismailia 41522, Egypt

[7] Department of Biology, College of Science, Princess Nourah bint Abdulrahman University, Riyadh 11474, Saudi Arabia; may_binjumah@outlook.com

[8] Department of Agronomy, Sher-e-Bangla Agricultural University, Dhaka 1207, Bangladesh; mhzsauag@yahoo.com

[9] Department of Electronics Engineering, Hellenic Mediterranean University, Chania, 73100 Crete, Greece; dkalderis@chania.teicrete.gr

* Correspondence: shafaqataligill@yahoo.com or shafaqataligill@gcuf.edu.pk

**Abstract:** Heavy-metal (HM) pollution is considered a leading source of environmental contamination. Heavy-metal pollution in ground water poses a serious threat to human health and the aquatic ecosystem. Conventional treatment technologies to remove the pollutants from wastewater are usually costly, time-consuming, environmentally destructive, and mostly inefficient. Phytoremediation is a cost-effective green emerging technology with long-lasting applicability. The selection of plant species is the most significant aspect for successful phytoremediation. Aquatic plants hold steep efficiency for the removal of organic and inorganic pollutants. Water hyacinth (*Eichhornia crassipes*), water lettuce (*Pistia stratiotes*) and Duck weed (*Lemna minor*) along with some other aquatic plants are prominent metal accumulator plants for the remediation of heavy-metal polluted water. The phytoremediation potential of the aquatic plant can be further enhanced by the application of innovative approaches in phytoremediation. A summarizing review regarding the use of aquatic plants in phytoremediation is gathered in order to present the broad applicability of phytoremediation.

**Keywords:** phytoremediation; heavy metal; aquatic plants; floating aquatic plants; wastewater treatment

## 1. Introduction

Water contaminations, along with limited availability of water, have put a severe burden on the environment. Around 40% population of the world is facing the problem of water scarcity due to climate change, rapid urbanization, food requirement and unchecked consumption of natural resources [1,2]. During the past few decades rapid urbanization, industrialization, agricultural activities, discharge of geothermal waters and olive wastewater especially in olive-cultivating areas enhanced the discharge of polluted wastewater into the environment [3–6]. Wastewater carrying soaring concentrations of pollutants is immensely noxious for aquatic ecosystem and human health [7–9]. Reclamation of wastewater has been the only option left to meet the increasing demand of water in growing industrial and agricultural sectors [10].

Industrial and domestic untreated wastewater contains pesticides, oils, dyes, phenol, cyanides, toxic organics, phosphorous, suspended solids, and heavy metals (HMs) [11]. Heavy metals among these toxic substances can easily be accumulated in the surrounding environment [12]. Commercial activities such as metal processing, mining, geothermal energy plants, automotive, paper, pesticide manufacturing, tanning, dying and plating are held responsible for global contamination of heavy metals [13,14]. Removal of heavy metals from the wastewater is difficult because they exist in different chemical forms. Most metals are not biodegradable, and they can easily pass through different trophic levels to persistently accumulate in the biota [15,16].

Removal of toxic pollutants is extremely important to minize the threat to human health and the surrounding environment. Removal of heavy metals achieved through various techniques such as reverse osmosis [17], ion exchange [18], chemical precipitation [19], adsorption and solvent extraction [20] include enormous operational and maintenance costs and are usually not environmentally friendly [19–22]. These conventional techniques for the remediation of heavy metals are generally costly and time-consuming. These treatment technologies require high capital investment and in the end, generate the problem of sludge disposal [23]. For the remediation of wastewater polluted with heavy metals contaminants, an environmentally friendly and economical treatment technology is needed [24,25]. The current study illustrates an environment-friendly technique phytoremediation for removal of contaminants on a long term basis. Furthermore, this review article summarizes the potential application of aquatic plants in phytoremediation for the treatment of wastewater.

## 2. Heavy Metals in the Environment

Anthropogenic and geological activities are the main source of heavy-metal pollution. Activities such as mining, military activities, municipal waste, application of fertilizer, discharge of urban effluent, vehicle exhausts, wastewater, waste incineration, fuel production, and smelting cause the production of metal contaminants [26,27]. Natural sources of heavy-metal pollution include erosion, weathering of rocks and volcanic eruption. Parent material during weathering is the primary and initial natural source of heavy metals [28].

Agricultural pesticides and utilization of fertilizers on agricultural soil have raised the concentration of Cd, Zn, Cu and As in soil [29]. A constantly increasing need for agricultural produce has increased the application of pesticides, fertilizers, and herbicides. This excessive use of these agrochemicals may result in the accumulation of these pollutants in plants and the soil as well [30]. Usage of phosphate fertilizer and inorganic fertilizers to control the diseases of crops, grain and vegetable sometime hold an uneven level of Ni, Pb, Zn, Cd, and Cr [31,32]. An enormous quantity of fertilizers is applied to deliver the K, P and N in order to improve the growth of crops, which in turn increase the incidence of cadmium, lead, iron and mercury in substantial high concentrations. Inputs of heavy metal to agricultural land through the excessive use of fertilizers is increasing apprehension about their probable hazard to the environment [33,34].

Wastewater irrigation leads to the buildup of various heavy metals like cadmium, lead, nickel, zinc, etc. Some of these metals like Zn, Cu, Ni, Cd and Pb are frequently present in the subsurface of the soil irrigated with untreated wastewater. Wastewater irrigation for long periods of time increase the concentration of heavy metal in the soil at toxic levels [35]. The unregulated dumping of

municipal solid waste is also another main source of raised soil contamination load. Open dumps and land filling are the common practices using worldwide to dispose of municipal solid waste. Despite being a useful source of nutrients, these wastes are also a source of some harmful toxic metals as well. Precarious and overload applications of fertilizers, pesticides and fungicides are very important sources of metal pollution [36]. Metal contamination can also be caused by transportation. Maintenance and deicing operations on roads also generate groundwater/surface contaminants. Corrosion, tread wear, and brake abrasion are well-recorded sources of heavy metals generation linked to highway traffic [37].

## 3. Phytoremediation

Phytoremediation is considered an effective, aesthetically pleasing, cost effective and environmental friendly technology for the remediation of potentially toxic metals from the environment. Plants in phytoremediation accumulate contaminants through their roots and then translocate these contaminant in the aboveground part of their body [38,39]. The notion of using metal accumulator plants for the removal of heavy metals and several other contaminants in phytoremediation was first introduced in 1983, but this idea has already been implanted for the last 300 years [40]. Phytoremediation is known by different names such as agro-remediation, green remediation, vegetative remediation, green technology and botano remediation [4,41,42].

Use of vegetation, soil and micro biota along with other agrochemical practices makes the vegetative remediation an appealing green technology for the accumulation of different heavy metals [43,44]. The application of in situ and ex-situ remediation is applicable in a phytoremediation process. In situ application is used more commonly because it reduces the multiplication of contaminant in water and airborne waste, which ultimately minimize the risk to the adjacent environment [45]. More than one type of pollutant can be treated on site by the phytoremediation without the need for a disposal site. It also reduces the spread of contamination by preventing soil erosion and leaching [46]. The clean up cost of phytoremediation is far less than other conventional techniques of remediation, which is the utmost advantage of this technique [47]. Phytoremediation is a relatively straightforward technique as it does not require any highly specific personnel and exclusive equipment. This is applicable for the remediation of large scale area where other conventional techniques prove to be extremely inefficient and costly as well [48].

An enormous number of contaminants can be remediated by phytoremediation technology such as insecticides, chlorinated solvents, Polycyclic aromatic hydrocarbons (PAHs), Polychlorinated biphenyl (PCBs), petroleum hydrocarbons, radio nucleosides, surfactants, explosive elements and heavy metals [48,49]. There are a number of plant species that have the ability to accumulate significantly higher concentrations of heavy metals in different parts of the body, such as a leaf, stems and root, without showing any sign of toxicity [50,51].

### 3.1. Characteristics of Phytoremediation Plants

Plants should have the following characteristics in order to make the phytoremediation an eco-sustainable technology: native and quick growth rate, high biomass yield, the uptake of a large amount of heavy metals, the ability to transport metals in aboveground parts of plant, and a mechanism to tolerate metal toxicity [52–55]. Other factors like pH, solar radiation, nutrient availability and salinity greatly influence the phytoremediation potential and growth of the plant [51,56].

### 3.1.1. Mechanism of Phytoremediation

Phytoremediation follows different mechanisms such as phytoextraction, phytostabilization, phytovolatilization and rhizofiltration during the uptake or accumulation of heavy metals in the plant [4,41]. The different mechanisms involved in the phytoremediation process are briefly describe below.

Phytoextraction

Phytoextraction is also called phytoaccumulation, and it involves the uptake of heavy metal in the plant roots and then their translocation into an above ground-level portion of the plant like shoots, etc. Once the phytoextraction is done the plant can be harvested and burned for gaining energy and recovering/recycling metal if required from the ash [57,58]. Sometimes phytoremediation and phytoextraction are used synonymously, which is a misconception; phytoextraction is a cleanup technology while phytoremediation is the name of a concept [59]. Phytoextraction is an suitable phytoremediation technique for the remediation of heavy metals from wastewater, sediments and soil [52,60].

Phytostabilization

Phytostabilization involves the use of the plant to restrict the movement of contaminants in the soil. The term phytostabilization is also known as in place deactivation. Remediation of soil, sludge, and sediment can be effectively done by using this technology. It does not interfere with the natural environment and is a much safer alternative option [61,62]. In phytostabilization, plants inhibit or act as a barrier for the percolation of water within the soil. When we need to persevere in our surface water, ground water and restoration of soil quality, this technology is best suited for this purpose because it cuts short the movement of the contaminants [63,64]. Phytostabilization is very effective for a large site, which is heavily affected by the contaminants [65]. Phytostabilization is only a managing approach for inactivating/immobilizing the potentially harmful contaminants. It is not a permanent resolution, because only the movement of metals is restricted, but they continue to stay in the soil [66].

Rhizofiltration

Rhizofiltration involves the use of the plant to ab/adsorb the contaminants, resulting in restricted movement of these contaminants in underground water [67,68]. Roots play a very significant part in rhizofiltration. Factor such as changing pH in the rhizosphere and root exudates helps the precipitation of heavy metal on the surface of the roots. Once the plant has soaked up all the contaminants, they can easily be harvested and disposed [69]. Plants for rhizofiltration should have the ability; to produce a widespread root system, accumulate high concentrations of heavy metals, be easy to handle and have low maintenance cost [42,70]. Both aquatic and terrestrial plants with long fibrous root systems can be used in rhizofiltration [70]. Rhizofiltration is productively used for handling and treatment of the agricultural runoff, industrial discharge, radioactive contaminant, and metals [71]. Heavy metals which are mostly retained in the soil such as cadmium, lead, chromium, nickel, zinc, and copper can be adequately remediated through rhizofiltration [72].

Phytovolatilization

Phytovolatilization is the process in which a plant converts pollutants into a different volatile nature and then their successive release into the surrounding environment with the help of the plant's stomata [48,73]. Plant species like canola and Indian mustard are useful for the phytovolatilization of selenium. Mercury and selenium are the most favorable contaminants that can be remediated in phytovolatilization [74]. One of the greatest advantages of phytovolatilization is that it does not require any additional management once the plantation is done. Other benefits are minimizing soil erosion, no disturbance to the soil, unrequited harvesting, and the disposal of plant biomass [75]. Bacteria present in the rhizosphere also help in the biotransformation of the contaminant, which eventually boosts the rate of phytovolatilization.

*.2. Advances in Phytoremediation*

### 3.2.1. Chemical Assisted Phytoremediation

The phytoremediation potential depends upon the phyto-availability of different heavy metals present in the soil [76]. The application of specific chemicals has proved to be a successful technique to boost the bioavailability of heavy metals to plants [41]. Organic fertilizers and chelating reagents are commonly used to decrease the pH of soils, which ultimately enhance the bioavailability and bioaccumulation in plants. In tobacco, decreased pH by application of a chelating reagent showed increased accumulation of Cd. The application of ethylenediaminetetraacetic acid (EDTA) boosted the phytoextraction and bioaccumulation of Cd, Zn, and Pb in various studies [77,78]. Some other chelating agents, diethylene triamine penta-acetic acid (DTPA) and ethylene glycol tetra-acetic acid (AGTA), also have been proved efficient chelators to enhance the phytoavailability and phytoextraction of heavy metals [79]. Organic acids such as malic acid, acetic acid, citric acid and oxalic acid have been proved effective chelating agents. The phytoremediation potential of plants may also be enhanced by strengthening plants to tolerate heavy-metal stress and toxicity. Application of salicylic acid (SA), has been found effective to alleviate metal stress in the plant, resulting in enhanced phytoremediation potential of plants [80,81].

Application of different chemicals also has some drawbacks. The applied chemical may cause the toxicity in plants, may leach to groundwater, and may disturb the translocation of heavy metals in plants. The applied chemicals often may form complexes with heavy metals, which have non-biodegrade abilities, leading to a source of secondary pollution [82]. The application of chelators may disturb the plant growth and development. It may result in decreased growth of roots, shoot, and biomass due to the toxic effects of chelators [83]. The negative impacts of chelators can be minimized by the application of a proper amount of the chelators, cautious application, and proper understanding of the water seepage mechanism [84]. The organic acids have advantages over synthetic chelators being economical and easily biodegradable and environment-friendly [85,86].

### 3.2.2. Microbial Assisted Phytoremediation

Plant-associated microorganisms have a key role in the remediation of heavy metals from soils [87]. These microorganisms influence the availability and accumulation of heavy metals in soil and plants. Recently, bio-augmentation of plants with particular and adapted microbes has been extensively studied in phytoremediation [38,53,88]. Plant growth-promoting rhizobacteria (PGPR) proved to increase biomass production, disease resistance, and reduce metal induced toxicity in bio-augmented plants [89]. Similarly, endophytic bacteria also play a very prominent part in phytoremediation [90,91]. The plant–endophyte interaction, fortify the plants to tolerate both biotic and abiotic stress [92]. Endophytes have developed several mechanisms to alleviate metal toxicity in plants. These methods include efflux of toxic metal ions from the cell, the transformation of metal ions into less-hazardous forms, sequestration, precipitation, adsorption, and biomethylation [93]. Application of rhizospheric and endophytic bacteria in soils/plants improves plant growth and boosts the phytoremediation potential of plants by enhancing metals availability, metals uptake, accumulation, reduced metal stress in plants. Furthermore, the rhizospheric and endophytic bacteria also enhance the phytoremediation potential of plants by enhancing soil fertility by the production of growth regulators and the provision of essential nutrients [94–96]. The mycorrhizal fungi in the root zone form an association with the roots of plants, and have a beneficial role in phytoremediation [97]. This plant-fungi association enhance the availability of essential plant nutrients through their hyphal network, modify the root exudates, alter soil pH and stimulate the bioavailability of various heavy metals to associated plants [98,99].

### 3.2.3. Transgenic Plants

The application of transgenic plants in phytoremediation is a novel approach to enhance the effectiveness of phytoremediation. Specific genes in transgenic plants increase the metabolism,

accumulation and uptake of definite pollutants [94,100]. The ideal plant to engineer for phytoremediation should possess characteristics; high biomass yield adopted to local and target environment and well-established transformation protocol. Transgenic plants also enhance the detoxification process of organic pollutants and the addition of toxic compounds in the food chain [100,101]. Firstly, transgenic plants were introduced for the remediation of inorganic pollutants; now they are effectively used to remove organic pollutants from contaminated media [102]. *Nicotiana tabaccum* and *Arabidopsis thaliana* are an example of transgenic plants firstly practiced for effective removal of heavy metals, cadmium, and mercury, respectively [103,104]. Transgenic plants have been proved efficient for the treatment of phenolic, chlorinated, and explosives contaminants [105,106].

Plants can be engineered to degrade the organic pollutants in the rhizosphere. In this, transgenic plants do not uptake and accumulate the pollutants; rather, incorporated genes secrete enzymes which degrade organic pollutants in the rhizospheric zone [107]. This approach also solves the problem of plant harvesting and handling loaded with toxic metals, as all the metal detoxification and removal process occurs in the rhizosphere by roots [108]. The transgenic *Arabidopsis* plants enhanced the degradation of 2,3-dihydroxybiphenyl (2,3-DHB). Similarly, transgenic tobacco plants speed up the detoxification of 1-chlorobuatne in the rhizospheric zone [109]. This ability of transgenic plants is attributed to the increased diversity of the microbial community, increased metabolic activity, the release of root exudates and enzymes and increased contact between roots and contaminants [110,111].

### 3.2.4. Non-Living Plant Biomass

Non-living plant biomass can be profitably used for metal uptake and metal recovery. Successive use of dried and dead biomass of plants (as simple biosorbent substance) to remove the metals from water has gained popularity over the past few years because it is easy to handle and is a cost-effective natural approach [112,113]. Water hyacinth's (*Eichornia crassipes*) dried roots showed the potential to remove cadmium and lead effectively from wastewater [114,115]. Biomass of different aquatic plant species such as *Eichhornia crassipes*, *Potamogetonlucens*, and *Salvinia herzegoi* was reported to be successfully used as an exceptional biosorbent material for the removal of Cr, Ni, Cd, Zn, Cu, and Pb effectively in various studies [116,117].

## 4. Aquatic Plants and Phytoremediation

The aquatic ecosystem is a cost-effective and resourceful clean up technique for phytoremediation of a large contaminated area. Aquatic plants act as a natural absorber for contaminants and heavy metals [118]. Removal of different heavy metals along with other contaminants through the application of aquatic plants is the most proficient and profitable method [52,119]. Constructed wetlands along with aquatic plants were extensively applied throughout the world for the treatment of wastewater [120,121]. The selection of aquatic plant species for the accumulation of heavy metal is a very important matter to enhance the phytoremediation [71,122].

Over the years, aquatic plants have gained an overwhelming reputation because of their capacity to clean up contaminated sites throughout the world [120,123]. Aquatic plants always develop an extensive system of roots which helps them and makes them the best option for the accumulation of contaminants in their roots and shoots [124,125]. The growth and cultivation of aquatic plants are time-consuming, which may restrict the growing demand of phytoremediation [126]. Nevertheless, this shortcoming is substituted by the number of advantages that this technology possesses for the treatment of wastewater [100,127].

### 4.1. Types of Aquatic Plants

### 4.1.1. Free-Floating Aquatic Plants

These are the plants with floating leaves and submerged roots. Some of the free-floating aquatic plants are well recognized for their capability to eliminate the metals from the contaminated

environment: water hyacinth (*Eichhornia crassipes*) [128], water ferns (*Salvinia minima*) [129], duckweeds (*Lemna minor*, *Spirodelaintermedia*), [130,131], water lettuce (*Pistiastratoites*), [132], water cress (*Nasturtium officinale*) [133]. The potential of these free floating aquatic plant for the elimination of heavy metals is comprehensively studied in different studies [99,134,135]. Active transport of heavy metals in free-floating aquatic plants occurs from the roots, from where metals are transferred to other parts of the plant body. Passive transport is associated with the direct contact of the plant body with the pollution medium. In passive transport, heavy metals mainly accumulate in upper parts of the plant body [136]. Water hyacinth, duckweed and water lettuce are the most frequently used free-floating plants for the remediation of heavy metals from wastewater [137–140]. The aptitude of different aquatic plants to mitigate different heavy metals is mentioned in Table 1.

**Table 1.** Accumulation potential of various aquatic plants.

| Aquatic Plant | Common Name | Metals/Metalloids | Reference |
|---|---|---|---|
| *Eichhornia crassipes* | Water hyacinth | Pb, Hg, Cu, Cr, Ni, Zn. | Molisani et al. [141]; Hu et al. [142] |
| *Pistia stratiotes* | Water lettuce | Cr, Zn, Fe, Mn, Cu | Maine et al. [136]; Miretzky et al. [143] |
| *Salvinia minima* | water spangles | As, Ni, Cr, Cd | Olguin et al. [135]; Sooknah, [144] |
| *Salvinia herzogii* | Water fern | Cd, Cr | Maine et al. [136]; Sun˜e et al. [145] |
| *Lemna minor* | Duckweed | Cr, As, Ni, Cu, Pb | Kara [146]; Ater et al. [147]; Basile et al. [148] |
| *Spirodela intermedia* | Duckweed | Fe, Zn, Mn, Cu, Cr, Pb | Miretzky et al. [143]; Cardwell et al. [149] |
| *Nasturtium officinale* | Water cress | Cr, Ni, Zn, Cu, | Kara [146]; Zurayk et al. [150] |
| *Myriophyllum spicatum* | Parrot feathers | Pb, Cd, Fe, Cu | Sivaci et al. [151]; Branković et al. [152] |
| *Ceratophyllum demersum* | Hornwort | As, Cd, Cr, Pb | Bunluesin et al. [153]; El-Khatib et al. [154] |
| *Potamogeton crispus* | Pondweed | Cu, Fe, Ni, Zn, and Mn | Borisova et al. [155] |
| Potamogeton pectinatus | American pondweed | Cd, Pb, Cu, Zn | Singh et al. [156]; Penga et al. [157] |
| *Typha latifolia* | common cattail | Zn, Mn, Ni, Fe, Pb, Cu | Hejna et al. [158]; Qian et al. [159]; Sasmaz et al. [160] |
| *Mentha aquatica* | Water mint | Pb, Cd, Fe, Cu | Branković et al. [152]; Kamal et al. [161] |
| *Vallisneria spiralis* | Tape grass | Ar | Giri [162] |
| spartina alterniflora | Cordgrass | Cu. Cr, Zn, Ni, Mn, Cd, Pb, As. | Aksorn and Visoottiviseth [163]; Hempel et al. [164] |
| *Phragmites australis* | Common reed | Fe, Cu, Cd, Pb, Zn | Ganjalia et al. [165]; Ha and Anh, [166] |
| *Scirpus* | Bulrush | Cd, Fe, Al. | Kutty and Al-Mahaqeri [167] |
| Polygonum hydropiperoides | Smartweed | Cu, Pb, Zn | Rudin et al., [168] |

### 4.1.2. Water Hyacinth (*Eichhornia Crassipes*)

Water hyacinth (*Eichhornia crassipes*) is a free-floating aquatic plant which belongs to the family of Pontedericeae that is closely correlated with the lily family. Water hyacinth is the most widespread invasive vascular plant of the world. It has an extensive dark blue root system along with curved, straight leaves. The roots contain a stolon from which new plants are produced [169]. Water hyacinth possesses the unique ability to grow in heavily polluted environments and successively extract pollutants [134]. It has the advanced tendency of remediating different pollutants like organic material, heavy metals, total suspended solids, total dissolved solids, and nutrients [170–172]. Removal of nutrients and heavy metals are vastly reliant on the optimal growth rate of water hyacinth [169,173].

Water hyacinth (*Eichhornia crassipes*) is recommended to treat industrial wastewater, domestic wastewater, sewage effluents, and sludge ponds because it has (1) high absorption rate of different organic and inorganic contaminants (2) can tolerate an extremely polluted environment and (3) has a gigantic production rate of biomass [174]. *Eichhornia crassipes* has greater ability to remediate contaminants like arsenic, zinc, mercury, nickel, copper and lead from industrial and domestic wastewater streams [175–177].

Water hyacinth's derived ash and activated carbon showed good accumulation capacity of different HMs like cooper, nickel, zinc and chromium. It also holds the benefit of having the least

biological sludge production and creation of bio-sorbent, which facilitate metal recovery [178]. Major industries like paper, food processing, textile, leather, cosmetics, and dyes manufacturing results in the release of dye contaminants into the environments. Dyes are most stable and stand firm against oxidizing agents, which in the end enhance water pollution. The widespread root system and tolerance against these dyes help water hyacinth to effectively accumulate the reactive dyes [114,179]. Water hyacinth shows significance removal efficiency for Cd, Pb, Cu, Zn, Fe, As, Mn, Cr, As, Al and Hg as reported in different recent studies [180–183]. Shoot powder of water hyacinth removed Cr and Cu by 99.98% and 99.96% when exposed to tannery effluents [184]. Recent research studies conducted to check the removal efficiency of water hyacinth for heavy metals are given in Table 2.

**Table 2.** Recent studies on uptake of heavy metals by water hyacinth.

| Metals/Metalloids | Results | Conditions | Reference |
|---|---|---|---|
| Ni | Concentrations of Ni Areal parts-(0.29 ± 0.02 mg/kg) Roots-(3.34 ± 0.26 mg/kg) | 1, 2, 3 and 4 mg $L^{-1}$ concentration of nickel. | González et al. [24] |
| Cd | Initial concentration of cadmium was 0.3 while Cd in leaves of the plant was 31 ± 3. | Cadmium exposure at 1000 and 130 ug/L. | Shuvaeva et al. [180] |
| Al, Pb, AS, Cd, Cu | Removal rate: Al-(73%) Pb-(73%,) As-(74%) Cd-(82.8%) Cu-(78.6%) | Wastewater from steel effluents | Aurangzeb et al. [181] |
| Cd, Hg, Pb, Ni | Removal rate: Cd-(97.5%) Hg-(99.9%) Pb-(83.4%) Ni-(95.1%) | Initial concentrations of Cd: 0.24, Hg: 4.971, Pb: 1.199, Ni: 3.34 in industrial wastewater | Fazal et al. [182] |
| Cr, Cu | Tannery effluents | Removal rate. Cr-(99.98%) Cu-(99.96%) | Sarkar et al. [184] |
| Cd, Zn, Cu, Pb | Removal rate: Cd-(98%) Zn-(84%) Cu-(99%) Pb-(98%) | Anaerobic packed bed reactors system | Sekomo et al. [185]. |
| Cr, Zn | Removal efficiency of Cr. (63%) on 3rd day, (80%) on 9th day Removal efficiency of Zn. (67%) on 9th day, (96%) on 12th day, (100%) on 15th day. | Stock solutions | Swarnalatha and Radhakrishnan [186] |
| Pb, Cu, Mn, Cd | Uptake in leaves Pb-(3.40–5.06 mg/kg) Cu-(6.41–13.5 mg/kg) Mn-(62.9–67.9 mg/kg) Cd-(0.037–0.13 mg/kg) | Wastewater from mining. | Prasad and Maiti [187] |
| Mo, Ag, Ba, Pb, Cd | TF Mo-(0.85 ± 0.14) Ag-(0.18 ± 0.04) Ba-(0.12 ± 0.03) | Gold mine waste water. | Romanova et al. [188] |

| | Pb-(0.06 ± 0.01) | | |
| | Cd-(0.65 ± 0.09) | | |
| Zn, Cd, Cu, Pb | Removal rate:<br>Zn-(93.5%)<br>Cd-(95.16%)<br>Cu-(58.23%)<br>Pb-(98.33%) | Stock solutions. | Li et al. [189] |

### 4.1.3. Water Lettuce

Water lettuce (*Pistia stratiotes* L.*)* fit in the aracae/arum family. Water cabbage, Nile cabbage, water lettuce, jalkhumbhi, and shellflower are some of the other common names of these plants. It is mostly found in lakes, stream, and ponds [190]. *Pistia stratiotes* have 20 cm-wide and 10–20 cm-long pale green leaves. Whitish hair covers the lower surface of the plant. It has underwater hanging structure underneath the floating leaves [191]. Water lettuce possesses extraordinary tolerance over an extensive range of pH and temperature [192]. Extension and proliferation of water lettuce occur with the production of daughter plants. *P. Stratiotes* also produces seeds which remain present in water; their germination occurs during the wet seasons [193].

Water lettuce (*P. Stratiotes*) is an excellent contender for the phytoremediation of contaminants as it is more prone than other aquatic vegetation [194,195]. The plant has the capacity of reducing/removing nutrients such as biological oxygen demand (BOD), chemical oxygen demand (COD), dissolved oxygen (DO), pH, total Kjeldahl nitrogen (TKN), ammonia ($NH_3$), nitrite ($NO_2^-$), nitrate ($NO_3^-$) and phosphate ($PO_4^{3-}$), from drinking and surface water, storm water, sewage water and industrial wastewater [196–198]. The small size of *P. Stratiotes* by contrast with water hyacinth showed better removing capacity for the Zinc and mercury from industrial wastewater stream [199].

Uptake of Cu, Zn, Fe, Cr and Cd does not have any harmful effect on the plant which makes *P. stratiotes* eligible to be used as a hyperaccumulators plant for the mitigation of organic contaminants and heavy metals from wastewater on a broad scale [200,201]. The biomass of *P. stratiotes* reduces more than 70% of zinc and cadmium from the contaminated solution during the experiment [202]. Water lettuce is an excellent accumulator of Pb, Zn, Cu, Cd, Mg, Fe, and Mn as reported by different recent studies given in Table 3.

**Table 3.** Recent studies on uptake of heavy metals by water lettuce.

| Metals/Metalloids | Condition | Results | Reference |
|---|---|---|---|
| Fe, Mn, Cr, Pb, Cu Zn, Ni, Co | Three sites of Al-Sero drain Giza, selected for the collection of plant and water sample. | High value of BCF and RP observed positive correlation exist between Fe and Cu with root and shoots of plant. | Galal et al. [71] |
| Cd | 20 to 50 g of plant applied in container having 10 L river water and cadmium exposure of 1000 and 130 (g/L) concentrations. | BCF-(1270 ± 250). Initial concentration of cadmium was 0, 3 while Cd concentration in leaves of the plants was 32 ± 3. | Shuvaeva et al. [180] |
| Pb, Cu | 120 g of plant applied in 10 litre of steel industry effluents. | Removal rate: Pb-(70.7%) Cu-(66.5%) | Aurangzeb et al. [181] |
| Fe, Mn, Na, Ni, Pb, Cr, Cu, Zn, Al, Ca, Cd, Co, K, Mg | Plants covered two storm water detention ponds | 50% accumulation of Ca, Co, Cd, Mn, Zn and Mg in roots. More than 50% absorbance by roots for Pb, Ni, Cu, Cr and Al. | Lu et al. [197] |
| Cd, Zn | Initial concentration of Zinc, 1.8, 18, 50, 79, 105 mg/L, initial concentration of Cadmium 0.01, 0.1, 1, 10 mg/L. | Removal rate. 70% reduction for both Zinc and Cadmium. | Rodrigues et al. [202] |
| As | Initial concentration of As applied 0, 5, 10, 15 and 20 μM. | High absorbance of arsenic observed in the roots of the plants. | Farnese et al. [203] |
| Pb (II) | Stock solution (2000 mg/L) | Removal of Pb (II) 96%. | Volf et al. [204] |

4.1.4. Duck Weed

Duckweed is a free-floating aquatic plant which floats on the surface of slow-moving and still water. This plant belongs to family Araceae but is frequently classified in subfamily *Lemnoideae*. This family of free-floating plant species consists of five genera such as (1) *Wolffia*, (2) *Wolffiella*, (3) *Spirodela*, (4) *Lemna*, and (5) *Landoltia*, having no less than 40 species recognized [205]. Duckweed is also known as a water lens. These are richly found in ditches, canals, and ponds; these are smaller and faster-growing plants on the earth. They can survive in high pH (3.5 to 10.5) and temperature 7 to 35 °C) [206].

The capability of duckweed plant to develop in polluted site with tremendous variation in pH, temperature, nutrient level makes them effective for use in phytoremediation [207]. Duckweed can eliminate a vast variety of different heavy metals, inorganic and organic contaminants, pesticides, nutrients arise from agricultural runoff, sewage, industrial and domestic wastewater [131,138,208]. Duckweed can easily inhibit the growth of algae and fungi in different ponds because it has the ability to cover the ponds due to its widespread high growth rate. They also diminish nitrogen from these ponds by taking up ammonia and denitrification [209]. Removal of the nutrient with the application of duckweed biomass will help to upgrade the quality of water and degradation of water ecology. Duckweed shows higher removal aptitude for chemical oxygen demand (COD), biological oxygen demand (BOD), total nitrogen (TN), total suspended solid (TSS) and NH$_3$-N from wastewater under favorable environmental circumstances [131,207].

Much higher elimination of different HMs such as As, Cr, Cu, Zn, Ag, Hg, Pb and Cd has been done through different species of duck weed including *L. minor*, *L. Trisulca* and *L. gibba* from wastewater [4,210,211] *S. polyrhiza*, *L. gibba* and *L. minor* examined for their remediation efficiency of boron, arsenic, and uranium. *Spirodelapolyrhiza* was investigated and found to be a good phyto-remediator of arsenic [212], *L. gibba* was found to be appropriate for the remedy of boron with a lower concentration of 2 mgL$^{-1}$ without any harmful effect on biomass [213]. It can also accumulate uranium (120%), boron (40%), and arsenic (133%) [214]. *L. minor* found to be an excellent contender for the remediation of arsenic [215]. In comparison with other macrophytes, duckweed is the most suitable plant for phytoremediation.

Use of duckweed for the remediation of nutrient pollutants and HMs from industrial and agricultural wastewater was reported in previous reports [216,217]. Several researchers have reported that duckweed (*L. minor*) could take up a huge concentration of heavy metals such as nickel, manganese, zinc, uranium, arsenic, and copper [218]. *Lemna minor L* shows an increase in chromium uptake percentage of 6.1%, 26.5%, 20.5%, 20.2% at a different exposure concentration of chromium stress [219]. Duckweed has the ability to conserve nature by acting as a hyperaccumulator plant in phytoremediation technology. Table 4 shows recent studies of heavy-metal uptake from wastewater by duckweed.

**Table 4.** Recent studies on uptake of heavy metals by duckweed species.

| Metals/Metalloids | Condition | Results | References |
|---|---|---|---|
| Cr | 0, 10, 100, 200 μM Cr concentration | increase in chromium uptake percentage by *L. Minor* 6.10%, 26.5%, 20.5%, 20.2% | Sallah-ud-Din et al. [219] |
| Cr, Pb | 2, 4, 10 and 15 mg/L concentrations with using lab water. | Removalrate Cr-(86.2%–94.8%) Pb-(91.0%–96.4%). | Abdallah [220] |
| Cd | 0.5, 1.0, 1.5, 2.0, 2.5, and 3.0 mg/L concentrations applied. | Removal rate Cd-(42%–78%) | Chaudhuri et al. [221] |
| Ar | Initial artificial concentrations of 0.5, | Removal of arsenic more than 70% at 0.5 mg/L on | Goswami et al. [222] |

| | 1.0, and 2.0 mg/L. | 15th day of experiment. | |
|---|---|---|---|
| Fe | Concentrations of 100% and 50%, for (7, 14, and 21 days) | Maximum recommend = 5 mg/L Fe at 7 days. | Teixeira et al. [223] |
| Co, Cu, Fe, Cd, Ni, Mo, Mn, Zn, Cr, Se | Mining wastewater in rich with selenium | Removal rate: Co-(87%) Se-(55%) 35–60% removal rate for rest of the heavy metals. | Flores-Miranda et al. [224] |
| Pb, Cd | Artificial by concentration of (2, 5 and 10 mg/L) | Removal rate (1) Pb-(98.1%) in 10 mg/L at 7 pH. (60.1%) in 2 mg/L at 9 pH. (2) Cd-(84.8 %) in 2 mg/L at pH 7. (41.6%) in (10 mg/L at pH 9. | Verma and Suthar [225] |
| Cu, Zn, Cd | Initial concentrations Cu-(4.10 mg/L) Zn-(4.30 mg/L) Cd-(7.30 mg/L) | Cu-(0.381 ± 0.021 mg/g) Zn-(0.557 ± 0.009 mg/g) Cd-(1.251 ± 0.041 mg/g) | Török et al. [226] |
| Cd, Cu, Pb, Ni | Municipal and industrial wastewater. | Removal rate Ni (99%) 80% removal percentage for rest of metals. | Bokhari et al. [227] |

### 4.1.5. *Salvinia* (Water Fern)

Water Fern *(Salvinia auriculata*is), a small free-floating macrophyte, is extensively scattered in aquatic ecosystems. It has the ability to reproduce quickly and have the ability to settle widespread colonies in areas in no time. *Salvinia* can double its population within around 3 to 5 days under suitable conditions [228]. A substantial growth rate, ease to handle, wide distribution and sensitivity to various noxious entities support the application of *Salvinia* for used as a bio-indicator of pollution index and for phytoremediation as well [229].

Water Fern *(Salvinia)* species, especially *S. natans,* are potentially used in phytoremediation as it has an enormous capacity for removal of HMs due to the rapid growth rate and tolerance to toxic pollutants [230,231]. It can effectively be used for the treatment of different kinds of wastewater and waste produced in the constructed wetland [232]. Roots of *Salvinia* have a higher rate of metal accumulation. Accumulation of metal in *S. natans* and *S. minima* reduces As, with increasing concentration of phosphate while heavy-metal uptake increase with the addition of sulfur [233].

The presence of favorable environmental conditions along with the existence of certain nutrients and chelators will determine the fate of *Salvinia* in the hyperaccumulation of heavy metals [234]. Among different species of *Salvinia*, *S. minima* shows high bioaccumulation factor (BCF) for the accumulation of cadmium and lead [235]. *S. minima* has successfully been used for the remediation of high-strength synthetic organic wastewater [236]. Different species of the *Salvinia* (water fern) are an excellent accumulator of Fe, Cd, Ni, Mn, Zn and Pb as reported by several studies, and their details are given in Table 5.

**Table 5.** Recent studies on uptake of heavy metals by *Salvinia* species.

| Metals/Metalloids | Condition | Results | Reference |
|---|---|---|---|
| Cd, Ni, Pb, Zn | Initial concentrations Cd-(0.03 mg/L)<br>Ni-(0.40 mg/L)<br>Pb-(1.00 mg/L)<br>Zn-(1.00 mg/L) | Removal rate:<br>Zn-(0.4046 mg/m$^{-2}$)<br>Ni-(0.0595 mg/m$^{-2}$)<br>Cd-(0.0045 mg/m$^{-2}$)<br>Pb-(0.1423 mg/m$^{-2}$) | Iha and Bianchini [129] |
| Zn, Cu, Ni and Cr | 15 mg/L initial concentration. 10 g biomass of five plants. | Removal rate:<br>Zn-(84.8%)<br>Cu-(73.8%)<br>Ni-(56.8%)<br>Cr-(41.4%) | Dhir et al. [231] |
| Ni | 0, 20, 40, 80, 160 M concentrations of NiCl2 | Accumulation of Ni 16.3 mg/g | Fuentes et al. [237] |
| Cu, Cr, Pb, Cd | Initial concentration<br>Cu-(1.092 ± 0.026)<br>Cr-(2.201 ± 0.0024)<br>Pb-(2.974 ± 0.018)<br>Cd-(0.251 ± 0.017) | After treatment<br>Cu-(2.035 ± 0.014)<br>Cr-(1.052 ± 0.022)<br>Pb-(1.924 ± 0.012)<br>Cd-(0.018 ± 0.018) | Ranjitha et al. [238] |
| Pb, Ni, Cu, Zn, Mn, Fe, Cr, Cd | coal mine effluents | Removal rate<br>Pb-(96.96%)<br>Ni-(97.01%)<br>Cu-(96.77%)<br>Zn-(96.38%)<br>Mn-(96.22%)<br>Fe-(94.12%)<br>Cr-(92.85%)<br>Cd-(80.99%) | Lakra et al. [239] |

### 4.2. Submerged Aquatic Plants

In submerged aquatic plants, leaves are the main part for metal uptake. Passive movement of the cuticle results in the absorption of heavy metals. Polyglalacturonic acid of the cell wall and negatively charged cutin and pectin polymers of cuticle results in the sucking inward of minerals. Movement of Positive metal ions takes place due to this inward enhanced charged density [59]. They have the ability to remove heavy metals from water and sediments [240–242]. Some of the famous submerged plants such as parrot feather (*Myriophyllum spicatum*), coontail or hornwort (*Ceratophyllumdemersum*), pondweed (*Potamogeton Crispus*), American pondweed (*Potamogetonpectinatus*), *Mentha Aquatica*, *Vallisneria spiralis* and water mint are well known for their ability to accumulate Zn, Cr, Fe, Cu, Cd, Ni, Hg and Pb [152,154,155,157,243].

### 4.3. Emergent Aquatic Plants

These plants are usually found on submerged soil where the water table is 0.5 m below the soil. Accumulation of HMs in emergent plants varies from plant to plant, they have the skill to bio-concentrate most of the metals in below ground-level roots from water and sediments, while some of the emergent plants, distribute the burden of metals in aerial parts as well. For example, smooth cordgrass (*spartinaalterniflora*) take up heavy metals in leaves [164], common reed (*Phragmites australis*) bears most of the heavy metal burden in the roots of the plant [166]. Sequestration and detoxification of heavy metals occur at the cellular level in these plants [244]. Cattail (*Typha latifolia*), bulrush (*Scirpus* spp.), common reed (*Phragmites*) and smartweed

(*Polygonumhydropiperoides*) are the best emergent aquatic plant that can effectively be used for the phytoremediation of several HMs like Cd, Fe, Pb, Cr, Zn, Ni, Cu [160,167,168].

## 5. Significance of Aquatic Plants for Phytoremediation of Wastewater

Phytoremediation of heavy metals with aquatic plants has gained significant consideration due to its elegance and cost-effectiveness [39,227]. The earlier worker has demonstrated that aquatic plants have the capability to eliminate HMs from different kinds of wastewater [140,225,245,246]. Aquatic plants remove heavy metals via absorption or through surface adsorption and integrate them into their system, and then accumulation them in certain bounded forms [247,248]. Effluents from wastewater mitigated through the aquatic plants, thus causing less harm to the surrounding environment. A wide array of aquatic plants like water hyacinths, *Salvinia* sp., water lettuce, giant duckweed, and *Azolla* sp. have displayed tremendous ability for the phytoremediation of numerous kinds of wastewater [249,250]. This review briefly describes the effectiveness of these aquatic plants for the remediation of different types of wastewater.

### 5.1. Phytoremediation of Municipal Wastewater

Municipal wastewater possesses significant risk for the aquatic environment as it is a main cause of heavy metal pollution. Zn, Cu, Ni, Pb and Hg are potentially more noxious metals and they may cause chronic and acute health effects, bioaccumulation and phytotoxicity [251–253]. Application of aquatic plants for the removal of heavy metals from municipal wastewater, sewage water, spillage areas, and other polluted sites has become a common practice and experimental technique [254,255]. Aquatic plants can be used as bio-accumulators as they have the ability to accumulate high concentrations of HMs in their biomass [256,257]. Root and shoot tissues of *Typha domingensis* showed maximum accumulation of Zn, Cd, Ni, Fe and Mn during the first 48 h of study, planted in pots filled with municipal wastewater. [258]. *L. gibba* was studied for the accumulation of arsenic, boron, and uranium from the municipal wastewater as an alternative removal method. Results revealed that U, As and B were rapidly absorbed by the plant during the first 2 days of a 7-day experimental study [214]. Two rooted macrophytes *Typha angustifolia* and *Phragmites australis* removed 14%–85% of heavy metals such as zinc, lead, arsenic, nickel, iron, copper, aluminum and magnesium from municipal wastewater in a hydroponic study [259]. Similarly, aquatic plants *Typha latifolia* and *Phragmites australis* showed excellent removal efficiency of heavy metals from the municipal wastewater. Both these aquatic plants showed higher removal rate for aluminum (96%), copper (91%), lead (88%) and zinc (85%) and slightly less removal rate for iron (44%), boron (40%) and cobalt (31%) [260].

### 5.2. Phytoremediation of Industrial Wastewater

Discharge of industrial waste into soil and water signifies a more critical threat to human health, living organisms, and other resources [261]. Phytoremediation, along with newly developed engineering and biological strategies, has facilitated the successful removal of HMs from industrial wastewater through both phytostabilization and phytoextraction [262]. Twelve aquatic plants were tested for their phytoremediation capability for different HMs originating from the industrial wastewater in Swabi district, Pakistan. Results demonstrated that these aquatic plants significantly removed heavy metals from industrial wastewater with excellent removal efficiencies: Cd (90%), Cr (89%), Fe (74.1%), Pb (50%), Cu (48.3%) and Ni (40.9%), respectively [263].

Southern cattail (*Typha domingensi*) showed maximum accumulation of zinc, aluminum, iron, and lead, especially in roots rather than leaves from the industrial wastewater pond. Rhizofiltration was found as dominant mechanisms, which explained the phytoremediation potential of *Typha domingensis* [264]. Promising aquatic plants showed better strength for the phytoremediation of the industrial effluents than other plants. Aquatic macrophytes *Marsileaquadrifolia*, *Hydrillaverticillata* and *Ipomeaaquati ca*showed much better accumulation potential and translocation factor (TF) value for HMs (Fe, Cr, Zn, Pb, As, Hg, Cd and Cu,) from the industrial effluents as compared to the

terrestrial plants *Sesbaniacannabina*, *Eclipta alba* and algal species (*Phormidiumpapyraceum*, *Spirulina platensis*) [265]. A high diversity of aquatic plants has the following advantages for remediation: high removal efficiency, better habitat, and distribution resilience [47,58]. *T. domingensis* is such a dominant aquatic plant species having a high tolerance to a toxic environment and proficient in accumulation of HMs. Maine et al. [266] also reported that *T. domingensis* showed much better survival and removal efficiency for iron, zinc, nickel, and chromium released from industrial wastewater of metallurgy plant over other higher diversified aquatic plants. Such a type of aquatic plant can be used on a large scale to study the long-term removal performance.

## 5.3. Phytoremediation of Textile Wastewater

Wastewater from the textile industries is considered as most polluted wastewater among other industrial sectors [267]. Printing and dyeing process of textile industry effluents produce both organic and inorganic contaminants. Heavy metals in textile effluents are more toxic as they are more dangerous for public health [268]. Mahmood et al. [269] investigated the feasibility of *E. crassipes* for the eradication of copper, chromium, and zinc from five different textile industries from Lahore district, Pakistan. *E. crassipes* effectively removed 94.78% Cr, 96.88% Zn, and 94.44% Cu from the industrial wastewater sample during the investigation period of 96 h. In another study, *E. crassipes* removed 94.87% of cadmium from the textile wastewater [114]. It is well documented that amongst different aquatic plants, water hyacinth is the superlative contender for the phytoremediation of textile industry effluents [270]. Aquatic plants *Pistia stratiotes*, *Azollapinnata*, and *Salvinia, molesta* were found very competent for the elimination of Fe, Cu and Mn at 25% concentration of the textile effluents [271]. A hairy root system of aquatic plants plays a vital part in the remediation of pollutants from wastewater in phytoremediation [272].

Roy et al. [273] investigated the remediation capability of three free-floating (*Eichhornia crassipes*, *Pistia stratiotes*, *Spirodelapolyrhiza*) aquatic plants for Cu, Pb, As and Cr from textile wastewater effluents. These macrophytes expressed an extensive uptake tendency for heavy metals, and *Eichhornia crassipes* was detected as the most competent contender in the remediation of HMs due to its fibrous widespread root system. Similarly, Ajayi and Ogunbayo, [274] also reported the effectiveness of water hyacinth in remediation of Fe, Cu, and Cd from textile effluents. High removal percentage (70%–90%) for various heavy metals such as copper, chromium, zinc, iron, and lead from textile wastewater was observed with the water hyacinth as reported by different researchers [269,275,276]. Duckweed (*Lemna minor*) also showed great potential for the removal of Cr, Zn, Pb, and Cd from the textile wastewater [185]. It has been reported previously that the application of an aquatic macrophytes treatment system (AMTS) is beneficial for the remediation of textile wastewater [277].

## 5.4. Phytoremediation of Mining Effluents

Mining activities harmfully affect the whole environment and put an incredible burden on local fauna and flora. The process of mining operations includes the discharge of an enormous amount of toxic effluents into the aquatic environment [278]. Effluents of mining activities hold a much higher concentration of different pollutants like calcium carbonate, TDS, TSS and heavy metals [279]. Heavy metals originating from the mining effluents are very persistent in nature and can easily accumulate in the soil, water, sediment and also have the ability to enter the food chain via bioaccumulation and assimilation thus affecting the health of human and animals [280]. Various methods have been developed around the world to remove the HMs. Phytoremediation is such a method that showed promising results in the successful remediation of heavy metals originated from the mining effluents by employing aquatic macrophytes [281].

As indicated by Sasmaz et al. [246], aquatic plants were very effective in the removal of remove HMs from mining effluents. Mishra et al. [282] explore the potential of three aquatic plants *Spirodela Polyrhiza*, *Eichhornia crassipes* and *Lemna minor* for the effective elimination of heavy metals. *Eichhornia crassipes* removed much a higher percentage of heavy metals than the other two macrophytes. *Eichhornia crassipes* eliminated Fe, Cr, Cu by 70.5%, 69.1%, 76.9% from the mining

effluents. Similarly, *Eichhornia crassipes* effectively removed Cr (VI) by 99.5% from industrial mine effluents in 15 days of the experimental period [283]. Aquatic plant *Limnocharis flava* significantly remove the Hg from the mining effluents in a pilot scale experimental study of 30 days [284]. Most widely used aquatic plants used in the phytoremediation of mine effluents are floating, submerged and emergent. Emergent plants usually promote the elimination of HMs from the mine effluents via collective processes like retention and uptake of heavy metals over their respective tissues [285]. The emergent plant used in the remediation of heavy metals from mine effluents include *Phragmites australis*, *P. australis*, *P. karka*, *P. australis and T. dominguensis* [286,287]. Floating aquatic plants cannot improve adsorption via the substrate, however, they promote adsorption process to the plant biomass. Successfully used floating aquatic plants in the treatment of mine effluents include *Salvinia natans*, *Pistia stratiotes*, *Eichornia crassipes* [239,288]. Submerged aquatic macrophytes like *Ceratophylum demersum*, *Cabomba piauhyensis*, *Egeria densa*, *Myriophylum spicatum* and *Hydrilla verticillata* are recommended to be used for the phytoremediation of mine effluents as they have shown outstanding ability to accumulate HMs in their whole body biomass [289,290].

*5.5. Phytoremediation of Landfill Leachate*

Landfilling and open dumping are the most common way of treating municipal solid waste (MSW) worldwide [291]. Leachate forms as a result of interaction among waste in landfill, water from the soil, and different types of other liquid contaminants disposed of in the landfill. Intermittent and non-uniform percolation of moisture content occurrs via solid waste in landfill, which eventually leads towards the generation of landfill leachate [292]. Generated landfill leachate if not properly managed, can easily lead towards numerous adverse health and environmental impacts [293]. One of the major constraint in the management of landfill leachate is the lack of effective treatment methods for the huge amount of landfill leachate generated worldwide [294]. Different chemical and physicochemical approaches have been used to eradicate pollutants from the leachate. Unfortunately, these methods are generally expensive and complicated as well. Economically viable and environmentally friendly option is a priority in landfill leachate management. Jones et al. [295] reported that plant-based remediation technology i.e., phytoremediation is very successful in the treatment of landfill leachate. Aquatic plants such as *Gynerium sagittatum (Gs)*, *Colocasia esculenta (Ce)*, *Heliconia psittacorum (He)* have shown tremendous phytoremediation potential for the remediation of landfill leachate [296].

Aquatic plants have the ability to withstand the high pollution load of the landfill leachate without showing any sign of a significant cutback in biomass and growth rate [297]. *Eichhornia crassipes* in a floating system has shown the tremendous capacity of removing diverse HMs such as Cu, Ni, Pb, Cd and Cr from the landfill leachate, thus reducing the pollution density of the landfill leachate [298]. Many researchers have utilized *Eichhornia crassipes* for the successful eradication of contaminants from the landfill leachate [299,300]. Ugya and Priatamby [301] also reported high removal efficiencies for various heavy metals form the landfill leachate generated from the landfill site with the assistance of *Pistia stratiotes*. Application of Duckweed (*Lemna minor*) also showed a significant reduction in copper, zinc, lead, nickel, and iron from landfill leachate [131].

Extensive root systems of aquatic plants enhance the capacity of these aquatic macrophytes to extract large concentration of HMs via the root system and then transport them to aboveground parts of the plant body [302]. The depth of root zone in aquatic plants is a vital impediment in successful phytoremediation of landfill leachate. Extensive root systems of the aquatic plants serve as main entry route of heavy metals, and these plants mostly store these HMs in roots then leaves and stem as observed in earlier reports [282,303]. Grisey et al. [304] reported that *P. australis* accumulates large concentration of HMs from landfill leachate in its root zone more resourcefully than other part of the body. Similarly *P. Cyperus Papyrus* also showed maximum accumulation of lead in its root than shoots during the treatment of the Kiteezi landfill site, Uganda [305]. Parallel outcomes were also examined by [306]. Thus it is highly recommended that planting of aquatic plants should be promoted for the treatment of landfill leachate in order to avoid seepage of heavy

metals and other contaminants from landfill leachate onto aquifer to contaminate the water bodies during ultimate discharge and runoff of leachate [301].

## 6. Role of Aquatic Plants in Constructed Wetlands

Remediation of wastewater through constructed wetland has been magnificently executed over the last few decades worldwide as an appropriate management choice for wastewater [307]. Constructed wetlands (CWs) are designed to treat distinct form of wastewater within the controlled environment. A broad range of wastewaters such as agricultural [308], municipal [30], landfill leachate [296], storm water [310] and industrial wastewater [311] can be remediated in constructed wetlands. The constructed wetland provides a comparatively simple and cheap solution for controlling water contamination without disturbing resources of natural wetlands [312]. Aquatic plants are an imperative constituent in CWs for the remediation of wastewater. Aquatic plants in CWs have two significant indirect functions: (1) leaves and stem of the aquatic macrophytes ehance the surface area for significant attachment of microbial communities, (2) aquatic plants have the aptitude to transport gases like oxygen down towards the root zone to allow their roots to subsist in the anaerobic environment [313]. Rhizosphere excessively support the microbial communities that handle the necessary alteration of metallic ions, different compounds, and nutrients [314]. Therefore, the application of aquatic macrophytes in CWs helps in the remediation of wastewater polluted with different contaminants and also acts as a sink for the contaminants [285].

Elimination of heavy metals in CWs depends on the kind of metallic elements, their ionic form, season, substrate condition, and kind of plant species [315]. Dense population of aquatic plants in CWs considerably increased the effectiveness of HM remediation from the wastewater [316,317]. Aquatic plants play a precise and vigorous part in maintaining the biochemistry of wetlands via their active and passive circulation of essential ingredients [318]. Heavy-metal concentration in wetland aquatic macrophytes generally decreased in the following order: root > leaves > stems [319,320]. However, the concentration of heavy metals does not deliver sufficient evidence regarding the uptake of heavy metals in aquatic plants in wetlands. In wetlands, uptake of heavy metals depends heavily upon the biomass of the particular aquatic plant [321]. *E. crassipes* is one such plant which has the capability to double its biomass within a few days under favorable conditions. Most recently, Rai [322] reported the water hyacinth (*E. crassipes*) as the most appropriate wetland plant for the phytoremediation of metals from wastewater. Use of *E. crassipes* in the constructed wetland to remove heavy metals has been recommended as the best choice in order to make use of *E. crassipes* (nuisance weed) effectively throughout the world. Sukumaran et al. [323] reported the utilization of *E. crassipes* for the phytoremediation of Cd, Pb, Cu, Ar from industrial discharge by applying constructed wetland technology. *E. crassipes* showed much higher remediation potential for Cu, Ni, Fe, Cd, Zn, Cr than the other two free-floating aquatic plants *Pistia Stratiotes*, *Spirodela polyrhiza* during a 15-day experiment.

Ladislas et al. [324] indicated the accumulation of Cd, Zn and Ni in aquatic floating macrophytes *J. effusus* and *C. riparia* growing in wetlands receiving storm water. The ratio of HMs concentration was significant in roots then shoots. Dan et al. [325] examined the accumulation of different HMs such as Fe, Cd, Zn, Ni, Pb and Cr by *Juncus effuses* and *Phragmites australis* from landfill leachate through a lab-scale constructed wetland. Both aquatic plants showed much higher removal efficacy for the targeted metals. Similarly, Leung et al. [285] also reported high removal percentages for heavy metals with three aquatic plants (*Phragmites australis*, *Cyperus malaccensis* and *Typha latifolia*,) in CWs receiving wastewater from the mining industries. The phytoremediation potential of three aquatic wetland plants i.e., *Cyperus alternifolius*, *Cynodon dactylon* and *Typha latifolia*, was examined for the transfer and translocation of HMs from the root zone to upper parts of the body in a constructed wetland receiving refinery wastewater. Results affirmed that the highest concentration of Cr, Zn, Cd, Pb and Fe were accumulated by roots of the plants followed by leaves and stem [326]. Similarly, *Typha latifolia* showed maximum removal efficiency of 96%, 95% and 80% for Cd, Cr and Pb correspondingly in a laboratory-scale constructed wetland unit. [327]. A CW with *Phragmites australis* was assessed for the phytoremediation of municipal wastewater. Most of the

metals were significantly removed from the municipal wastewater with reasonable efficiencies. Results demonstrated that *Phragmites australis* accumulated most of the heavy metals in their belowground part, and only a minor fraction of metals translocated two aboveground biomass of the plant [328]. Vymazal et al. [329] observed a maximum amount of HMs usually found in belowground biomass, while the lowest concentration of HMs were detected in aboveground biomass of the wetland plants.

Hadad et al. [330] also stated a similar trend of much higher accumulation of HMs in the root zone than upper parts of the plant. Plants frequently tolerate the high concentration of metals because they restrict the accumulation and absorption to the leaves upholding a constant and comparatively low concentration of HMs in aboveground parts of the plant. Research over the past decade has shown aquatic plants contribute significantly in the elimination of heavy metals through constructed wetland technology [266,318,331,332]. Discrepancies with the remediation of HMs through aquatic wetland plants in CWs might be attributed to various aspects including the type of wetland, inflow load of heavy metals and type of wetland plants. Nevertheless, the type of aquatic plant employed in the CW system is one of the main prominent element in the remediation of metals from wastewater [333]. Further investigation is needed to increase the removal efficiency of these aquatic plants within constructed wetlands.

## 7. Other Advantages of Aquatic Plants

The study of phytoremediation reveals that the aquatic macrophytes have the advantage over other plants in the remediation of heavy metals [71,131,257]. The widespread availability, rapid growth rate, high biomass, cost-effectiveness and tolerance to toxic pollutants make them the best suited, available phytoremediation plants. Purifications system using these aquatic plants have gained more attention worldwide because of their capacity to accumulate and remove of a persistent organic pollutant from water bodies [48,131].

Involvement of appropriate phytoremediation technology needs intervallic harvesting of the plant biomass in order to assimilate and confiscate heavy metals and nutrients from water bodies. Conversion of biomass into exclusive material is the significant factor in promoting this technique for the treatment of contaminants. An aquatic plant's biomass can latter on be used as animal feed, useful in the production of biogas and compost as reported in many studies [25]. Bio-sorption along with bioaccumulation of *Lemna minor* biomass was inspected which indicates its possible use as animal feed [131,334]. Aquatic plants possess sugar in the shape of starch, cellulose, and hemicelluloses. Carbohydrate hydrolysis of this fermentable sugar results in the production of lactic acid, ethanol, and other important products. Therefore, sugar present in aquatic plants is a new promising feature supporting their role in the eco-sustainable environment. Aquatic plants i.e., *Pistia stratiotes* and *Eichhornia crassipes,* have been reported to produce sugar during their enzymatic hydrolysis process [335]. Free-floating aquatic plants (*Azolla* spp., *Wolffia* spp., *Spirodela* sp. and *Duckweeds*) can be used as a food source for water bird. They also provide shelter for insect larvae and small mollusks. Fishes also use the mats of these plants as cover and use their shade for reproduction [336]. Aquatic plants can be efficiently used to improve the aquaculture for fishponds. Elimination of nitrogenous waste of aquatic plants is also an added benefit for their application in aquaculture, i.e., *Canna generalis* L., *Typha angustifolia. Echinodoruscordifolius*, and *Cyperus involucratus* removed ammonia, nitrate and nitrite efficiently [337].

## 8. Conclusions and Future Prospects

Heavy metals in our environment as a persistent pollutant needs absolute elimination for a completely remedial objective. Utilization of phytoremediation seems to be a less disruptive, economical and environmentally sound clean-up technology. Choice of appropriate plant is the most significant feature in phytoremediation. Aquatic plants perform very vibrant roles in the remediation of heavy metals from the polluted site with equal ease to other hyperaccumulator plants. Application of aquatic plants both in bioaccumulation (with living plant biomass) and bio-sorption (with dead plant biomass) can be done successfully for the eradication of heavy metals.

Comprehensive interaction, transport, and chelator activities regulate the storage and accumulation of heavy metals by the aquatic macrophytes. Genetic engineering enhances the accumulation and tolerance capacity of plants, which shows its exceptional application in improving the effectiveness of phytoremediation. In plants, at the molecular level, different extensive steps have been evaluated that favor the transgenic methods in order to plead with the changeover metal fraction of plants. Genetically engineered plants show high tolerance and metal uptake capacity and, as a result, gene manipulation has successfully been investigated in terrestrial plants, but, genetic engineering of aquatic plants to enhance their heavy-metal uptake capacity is in its preliminary phases.

Disposal of plant biomass later on, can be used for the production of biogas and also can be used as animal feed. The application of aquatic plants in phytoremediation like other conventional physical and chemical techniques does not require any post-filtration and can be effectively used to treat a large volume of polluted water and soil. Based on the present review, the benefits of using aquatic plants to treat contaminants are huge, because this technology does not only treat the contaminants but is cost-effective and visually pleasing as well as being advantageous for the sustainability of whole ecosystems.

**Funding:** The authors are thankful for the financial support from the Government College University, Faisalabad. This research work was funded by the Deanship of Scientific Research at Princess Nourah bint Abdulrahman University through the Fast-Track Research Funding Program.

**Acknowledgments:** The authors are thankful for the financial support from the Government College University, Faisalabad. This research work was funded by the Deanship of Scientific Research at Princess Nourah bint Abdulrahman University through the Fast Track Research Funding Program.

**Conflicts of Interest:** The authors declare that there is no conflict of interests regarding the publication of this paper.

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
