# Peer review of "Application of Floating Aquatic Plants in Phytoremediation of Heavy Metals Polluted Water: A Review"

_sustainability, doi:10.3390/su12051927_

Round 1
Reviewer 1 Report
Heavy metal toxicity is a widespread problem. Therefore, a review article can provide an organized view of metal phytoextraction by floating aquatic plants. The highlighted findings in this manuscript will provide easy and wide understandings regarding the metal phytoextraction by using this plant. The text is well written - in good English. The authors provide a rich source of the present literature dealing with the subject - this is what I appreciate. My comments are below
- Please write to the point, no need to write a definition of phytoremediation in detail.
- Please check the English throughout the article
- There are few typo errors in the text such please correct it
- Do not start sentences with abbreviations and mention full names along with abbreviations when they first appear in the text. Please check and correct in the whole manuscript
- Check for reference style in the text and in the reference list and made it according to Journal style
- Please write metals/metalloids in place of metals in Table headings.
- In Table 2 Cd. No dot after Cd same in another place, please.
Author Response
Reviewer # 1
Heavy metal toxicity is a widespread problem. Therefore, a review article can provide an organized view of metal phytoextraction by floating aquatic plants. The highlighted findings in this manuscript will provide easy and wide understandings regarding the metal phytoextraction by using this plant. The text is well written - in good English. The authors provide a rich source of the present literature dealing with the subject - this is what I appreciate.
My comments are below
1 Please write to the point, no need to write a definition of phytoremediation in detail.
Answer: Thanks for the valuable suggestion, needful is done for improvement in new manuscript file.
- Please check the English throughout the article.
Answer: English improved where needed in the manuscript.
- There are few typo errors in the text such please correct it.
Answer: Correction made in this regard in whole manuscript.
4 Do not start sentences with abbreviations and mention full names along with abbreviations when they first appear in the text. Please check and correct in the whole manuscript.
Answer: Necessary improvements is made in the manuscript.
- Check for reference style in the text and in the reference list and made it according to Journal style.
Answer: References are set according to journal style.
- Please write metals/metalloids in place of metals in Table headings.
Answer: Corrected as suggested.
- In Table 2 Cd. No dot after Cd same in another place, please.
Answer: Needful corrections are made in whole manuscript as suggested by the respected reviewer.
Reviewer 2 Report
Dear colleague and authors, from my personal experience in this topic, I do not have suggestion to improve your paper. As a review it is very rich, of course it is not original, but is a good work for researchers working on this field. Very useful and interesting is the paragraph 5. Significance of Aquatic Plant for Phytoremediation of Wastewater
Author Response
Reviewer # 2
Dear colleague and authors, from my personal experience in this topic, I do not have suggestion to improve your paper. As a review it is very rich, of course it is not original, but is a good work for researchers working on this field. Very useful and interesting is the paragraph 5. Significance of Aquatic Plant for Phytoremediation of Wastewater.
Answer: Thanks for the encouraging comments.
Thank you again.